# Insights into the Variations of Hao-Dependent Nitrifying and Nir-Dependent Denitrifying Microbial Communities in Ammonium-Graduated Lake Environments

**Ruojin Zhao [1,†], Yinyan Chen [1,†], Jin Qu [1], Peng Jin [2,*], Zhanwang Zheng [1,3,*] and Zhiwen Cui [3]**

[1]   The Key Laboratory for Quality Improvement of Agricultural Products of Zhejiang Province, The College of Agricultural and Food Sciences, Zhejiang A & F University, Hangzhou 311300, China

[2]   School of Environmental & Resource, Zhejiang A & F University, Hangzhou 311300, China

[3]   Zhejiang Shuangliang Sunda Environment co., LTD, Hangzhou 310000, China

*   Correspondence: jinpeng@zafu.edu.cn (P.J.); Zwzheng@zafu.edu.cn (Z.Z.)

†   These authors contributed equally to this work.

**Abstract:** Biological nitrification and denitrification play significant roles in nitrogen-associated biogeochemical cycles. However, our understanding of the spatial scales at which microbial communities act and vary is limited. We used gene-specific metagenomic PCR to explore changes in nitrifying and denitrifying microbial communities within pristine lake and its branches, where the ammonium and dissolved organic carbon (DOC) concentrations form a gradient. The biomarkers hydroxylamine oxidoreductase and nitrite reductase genes indicated that strong relationships exist between the diversities and community structures of denitrifiers and ammonium gradients. It showed that the *Nitrosomonas oligotropha* cluster dominates the nitrifying bacteria in low-nutrition environments, while a new *Nitrosomonas ureae* cluster accounted for nearly 80% of the nitrifying bacteria in high-nitrogen environments. The distribution and diversity of nirS/K-dependent denitrifiers in the various habitats were similar, but predominantly affiliated with unknown clusters. Moreover, the abundance of all the *hao* genes dramatically outnumbered that of *nir* genes. The relative abundance of *hao* was clearly higher during eutrophication (13.60%) than during oligotrophy (5.23%), whereas that of *nirS* showed opposite tendencies. Overall, this study provides valuable comparative insights into the shifts in nitrifying and denitrifying microbial populations in lake environments with ammonium gradients, suggesting that unique dominant denitrifiers probably play an important role in the nitrogen cycle.

**Keywords:** microbial diversity; hydroxylamine oxidoreductase; nitrite reductase; ammonium gradient; metagenomic; denitrifiers

## 1. Introduction

The global biogeochemical nitrogen cycle has undergone dramatic changes in the past few decades. In particular, anthropogenic processes, including agricultural, industrial, and domestic activities, have resulted in an unprecedented increase in environmental nitrogen release. Overloading terrestrial ecosystems with ammonium, nitrate, and nitrite as the principal nitrogen pollutants has serious negative effects on human health, biodiversity, and water quality [1]. The coupled nitrification–denitrification and nitrification–anaerobic ammonium oxidation (anammox) processes mediated by environmental microorganisms are known to play important roles in the transformation and biogeochemical cycling of nitrogen, and probably contribute to the removal of up to 50% of the external dissolved inorganic

nitrogen that enters the ecosphere [2,3]. Environmental conditions (i.e., carbon or nitrogen) can significantly influence the community structures and relative abundances of microorganisms [4]. The abilities of particular lineages of organisms to survive in specific environments may reflect relevant aspects of environmental nutrient availability [5,6]. The diversity and structures of bacterial communities in lakes, estuaries [7], sea coasts [8,9], and wastewater treatment plants [10–12] have been extensively investigated. Most studies of community bacterial phylogenies from single-source environmental samples have been based on the conventional analysis of 16S rRNA genes or Denaturing gradient gel electrophoresis (DGGE) analysis of specific genera [13]. However, the great divergence among different bacterial taxa is not directly related to the major environmental factors. Thus, it is important to investigate the variations and shifts in community composition regulated by the aquatic environmental factors [14,15].

To date, there has been few investigations available on the comparative analysis of microorganisms in lake environments where nutrients vary greatly within the same region, especially those involving nitrogen metabolisms [7]. The lake reservoir is influenced by the dynamic inflow tributaries adjoining residential areas, and may serve as a transitional zone from the eutrophication phylotypes of nitrifying and denitrifying bacteria to the oligotrophication ones. Therefore, a refined and targeted comparative description of the diversity of nitrifying and denitrifying communities in freshwater lakes with ammonia gradients is required in order to improve our understanding of the shifts in the nitrifying and denitrifying populations that occur with sharp differences in the environmental nitrogen status. This should also offer an opportunity to identify populations that are specific to those environments.

The objectives of the present research were to investigate the pronounced community shifts in nitrifying and denitrifying populations in different environments (where there are sharp differences in the nitrogen and organic carbon status). A gene-specific metagenomic targeted PCR approach [16] was employed to comparatively characterize the nitrifying and denitrifying microbial communities from three environmental waters with different hydrological features, trophic statuses, and nitrogen pollution levels (Table 1). We expected that differences in the structures of nitrogen-transforming communities would accompany differences in nitrogen availability among three samples. We tested this by sequencing the hydroxylamine oxidoreductase (HAO) and nitrite reduction (copper-containing NirK and cytochrome cd1-containing NirS)-encoding genes of representative nitrifying and denitrifying bacteria within these samples, and compared their relative abundances with qPCR. These findings should extend our understanding of the bacterial communities that inhabit aquatic ecosystems and may have utility in the sustainable management of water resources.

**Table 1.** General physicochemical parameters of three environmental waters of the pristine Qingshan Lake and its branch, the Sewage River.

|  | TN | $NH_4^+$-N | $NO_3^-$-N | $NO_2^-$-N | COD | pH | DO |
|---|---|---|---|---|---|---|---|
| QL-H | 0.21 | 0.08 | 0.06 | ND [a] | 13 | 6.83 | 4.91 |
| QL-V | 16.82 | 14.36 | 1.15 | 0.14 | 75 | 6.97 | 5.71 |
| SR-W | 121.32 | 93.41 | 5.67 | 0.87 | 331 | 7.71 | 3.35 |

[a] ND not detectable, unit mg/L.

## 2. Materials and Methods

### 2.1. Sample Collection and Chemical Analyses

The Qingshan Lake reserve watershed area is approximately 620 km$^2$, and is the important potable water sources of Lin'an city. There are mainly six tributaries adjoining the residential area and inflowing to the Western basin of the lake. Three samples representing the three main nutrient levels (i.e., ammonium; total nitrogen, TN; and chemical oxygen demand, COD) were collected in Qingshan Lake from the tributary water to the reservoir area. All the sludge samples were collected from the surface sediments (5 cm) at a water depth of 30 to 50 cm. High-quality water samples of Qingshan

Lake (QL-H) and those closest to the village of Qingshan Lake (QL-V) were taken from a reservoir area and the adjoining residential area, respectively. The other Sewage River sample (SR-W) was taken from a domestically polluted tributary. A sample was taken by collecting at least six individual samples at the same depth in the same area to form a composite sample in the study. Metagenomic DNA was isolated from the sludge samples with the Mo Bio PowerSoil® DNA Isolation Kit (Mo Bio Laboratories, Inc., Carlsbad, CA, USA). To remove any coextracted humic substances and other contaminants, the metagenomic DNA was further purified and concentrated with ethanol precipitation, as previously described [17].

The dissolved oxygen (DO) concentrations were measured with the Winkler method [18]. The concentrations of ammonium, nitrate, and nitrite were determined with Nessler's reagent spectrophotometry at 420 nm, *N*-(1-naphthalene)-diaminoethane spectrophotometry at 410 nm, and ultraviolet (UV) spectrophotometry at 540 nm, respectively. Total nitrogen (TN) was measured with the standard UV spectrophotometry method (DR6000, Hach, Loveland, CO, USA).

## 2.2. Primer Design and PCR Amplification

Over 500 nucleotide and protein sequences of the hydroxylamine oxidoreductase (HAO) and nitrite reductase (NirS/NirK) genes were retrieved from the NCBI database (https://www.ncbi.nlm.nih.gov/protein/) with an identity >30% and size >500 b.p. Then, all the collected protein sequences were performed to multiple sequence alignment analysis and phylogenetic tree construction using the online tool Clustal Omega (https://www.ebi.ac.uk/Tools/msa/clustalo/). Then, some representative sequences from the sub-branch were selected to design gene-specific degenerate primers and phylogenetic analysis. Primers JBHAO-170F (GTATGAVGCGYTGGTNAAGCGYTA), JBHAO-939R (TGGAACTGGRAHGTHCVTCTCAAG), JBNirS-1001F (CGTGGTGGGAAAYTAYTGGCCKCC), JBNirS-1242R (CAYGAYGGHGGHTGGGAC), JBNirK-526F (CACGACGCTCACGGNATGTAYGG), and JBNirK-1002R (GTGCGCGACARCGCRTGRTCNAC) were designed according to the sequence alignments of the typical candidate *hao* (12), *nirS* (20), and *nirK* (19) nucleotide sequences, respectively.

Metagenomic DNA was used for the PCR amplification of 16S rRNA genes (prokaryote specific primers 27F (AGAGTTTGATCCTGGCTCAG) and 1492R (TACGGCTACCTTGTTACGACTT)), *nirS*, and *hao* genes. PCR products were separated by agarose gel electrophoresis and cloned into the pMD19-T vector (TaKaRa, Bio Inc., Shiga, Japan) according to the manufacturer's instructions. Randomly selected clones from each sample were sequenced by Shanghai Sangon Biological Engineering Technology & Services Co. Ltd., China.

## 2.3. Quantitative PCR Analysis of hao, nirS, nirK, and 16S rDNA Genes

To determine the gene abundances in 1 ng of extracted DNA, quantitative PCR (qPCR) was performed in a Corbett Real-Time PCR Machine with the Rotor-Gene 6000 series software 1.7 (Qiagen, Netherlands), using the SYBR Green. The abundances of nitrifiers and denitrifiers were measured by quantifying the *hao*, *nirS*, and *nirK* genes. The abundances of microbial communities were determined by quantifying the 16S rRNA gene. Standard curves were constructed for the *hao*, *nirS*, and *nirK* genes using the positive recombinant plasmids Hhao-05/pMD19, HnirS-4/pMD19, and VnirK-15/pMD19 as standards, respectively. For the 16S rRNA gene, the positive recombinant plasmid W16S-1/pMD19 was used as the standard. The samples and standards were analyzed with qPCR in triplicate, and the specificity of the qPCRs was confirmed with a melting curve analysis, agarose gel electrophoresis, and DNA sequencing. The efficiencies of the PCRs were 86.9–98.7%, with $R^2$ values >0.993 for all the calibration curves.

## 2.4. Phylogeny Analysis

Positive clones were randomly selected and sequenced using universal primers M13. DNA sequences were analyzed using the BLASTN tool, and amino acid sequences were analyzed using BLASTP (https://blast.ncbi.nlm.nih.gov/Blast.cgi) to aid the selection of the closest reference sequences.

To determine the evolutionary relationship of these environmental Nir and HAO enzymes with established nitrifying and denitrifying microbial communities, the sequences were compared to representative Nir and HAO enzymes from the non-redundant (NR) protein sequence database (NCBI), using neighbor-joining phylogenetic analysis. One thousand bootstrap replications were performed using the MEGA software (MEGA 5.0) [19].

### 2.5. Nucleotide Sequence Accession Numbers

Partial 16S rRNA gene sequences were deposited under genbank Accession No MH155326-MH155426, while partial *hao*, *nirS*, and *nirK* gene sequences were deposited under Accession No MH156255-MH156333, MH156334-MH156390 and MH156391-MH156420, respectively.

## 3. Result

### 3.1. Environmental Characteristics

To determine the relevance of the inorganic nitrogen concentration in the water to nitrifying and denitrifying microorganisms, the physicochemical characteristics of the samples collected in this study were determined and summarized in Table 1. The collected samples showed significant differences in their concentrations of TN, ranging from 0.21 to 121.32 mg/L. The TN concentrations were significantly higher in the polluted river samples than in the lake samples. Overall, ammonium, nitrate, nitrite, and the chemical oxygen demand (COD) showed distinct gradients, decreasing dramatically lakeward, with ranges of 93.41 to 0.08 mg/L, 5.67 to 0.06 mg/L, 0.87 to below the detection limit, and 334 to 13, respectively. The river sediments had a higher pH (7.71) than the lake sediments. The DO concentrations ranged from 3.35 to 5.71 mg/L in the three environmental points.

### 3.2. Distributional Differences in Microbial Communities in Nitrogen-Related Environments

Analysis of the 16S rRNA gene sequences in the genomic DNA from Qingshan Lake (QL-H and QL-V) and Sewage River (SR-W) are shown in Figure 1. A total of 108 16S rRNA gene sequences were obtained from the environmental sediment samples collected in this study, using 16S rRNA gene-based biomarkers and PCR amplification. A BLAST sequence analysis confirmed that approximately 93.52% of the amplified sequences (101 sequences) belonged to uncultured bacteria, and that they shared sequence identities of 88–100% with previously reported 16S rRNA sequences. A multiple alignment of all 108 sequences showed that the percentage identities of the nucleotide sequences were 29.14% to 100%. According to phylogenetic and statistical analyses, the 16S rRNA gene sequences were grouped into five distinctive clusters (Figure 1). From this unique dataset, we identified 17 different phyla, with the overwhelming majority of sequences (~90% of all sequences) belonging to a few bacterial phyla: Proteobacteria, *Aquabacterium*, Bacteroidetes, Cytophagales, and unclassified bacteria. More than 90% of these were uncultured microorganisms, and unclassified bacteria constituted approximately 40–60% of the sequenced clones. A phylogenetic analysis showed that Proteobacteria comprised 35% of the total sequences, and comprised an even larger proportion in the 16S rRNA metagenomic clone library, which might truly reflect the community diversity. Proteobacteria was the dominant phylum in all three sludge samples, accounting for about 39.40%, 25.81%, and 43.18% of the phyla in samples QL-H, QL-V, and SR-W, respectively (Figure 1).

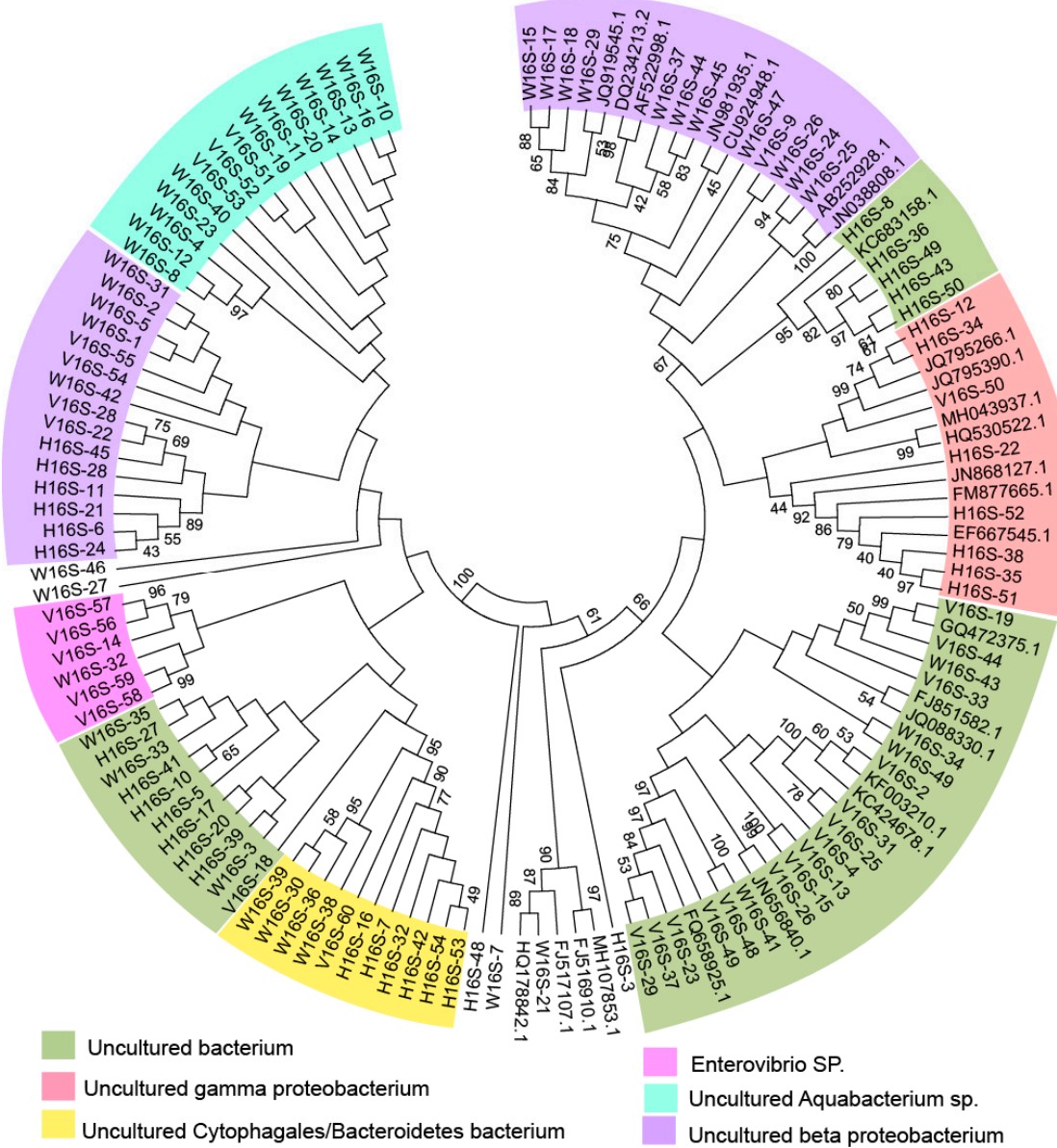

**Figure 1.** Analysis of microbial communities based on 16S rRNA partial gene sequences retrieved from three environmental collections and other representative organisms' species. Clones H16S, V16S, and W16S were from a high-quality Qingshan Lake water site (QL-H), the Qingshan Lake site closest to the village (QL-V), and domestic pollution in the Sludge River (SR-W), respectively.

*3.3. Phylogenetic Diversity of Hao-Dependent Nitrifying Bacteria in Sediments*

The nitrifying bacterial *hao* genes in the three samples were amplified from the metagenomic DNA (template) with primers JBHAO-170F and JBHAO-939R to detect the presence of ammonia-oxidizing bacteria (AOB). The PCR products revealed the presence of AOB in these samples (Figure 2). The amplification products were cloned and sequenced, and three clone libraries were constructed. An analysis of randomly selected clones indicated that all the clones of approximately the expected size (817 b.p.) were *hao* gene fragments, and no non-*hao* genes were amplified. A total of 132 clones from the three clone libraries were randomly selected and sequenced. The BLASTP results indicated that all the clones contained highly similar *hao* sequences, with approximately 90–99% identity to other *hao* gene sequences in the database. Only 79 novel gene sequences were detected among the sequenced clones, except for the redundant clone sequences. The deduced amino acid sequences of these gene fragments

matched the conserved regions in the previously characterized functional domains within other Hao proteins. These results demonstrate that the PCR primers that were newly designed to detect *hao* genes in various sediment samples, JBHAO-170F and JBHAO-939R, are useful for identifying the diversity of AOB nitrifiers with high resolution (100% coverage) and specificity. A multiple alignment of all these sequences showed that the percentage identity between pairs of Hao protein sequences ranged from 82.96% to 99.63%. The nucleic acid sequences corresponding to these protein sequences showed between 77.90–99.63% identity.

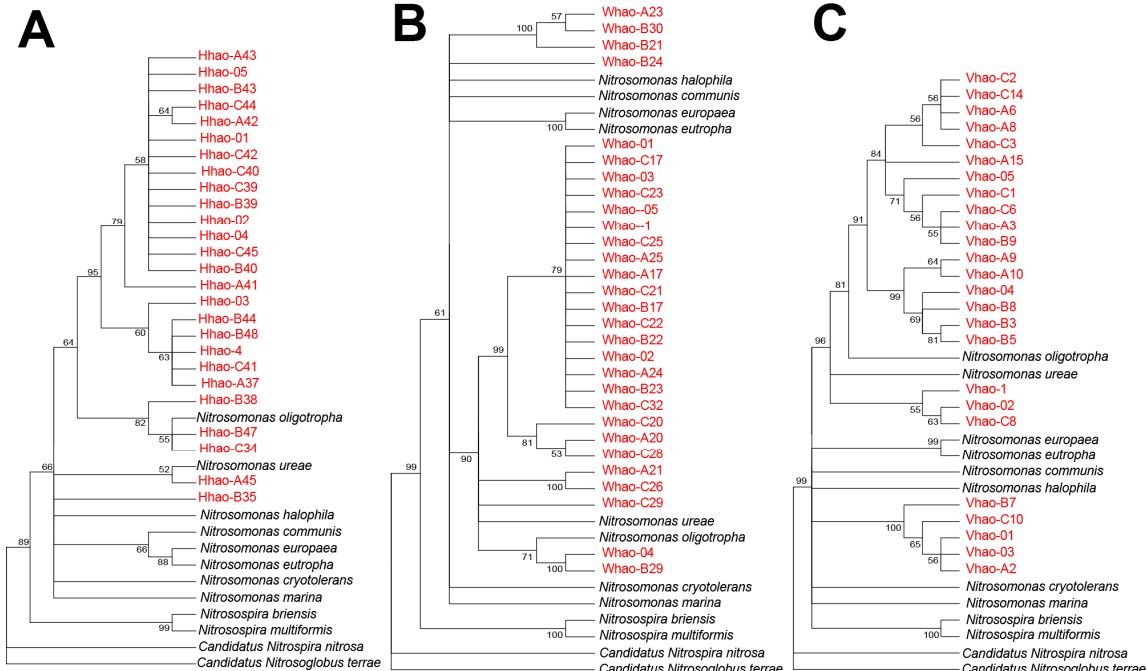

**Figure 2.** Trees of hydroxylamine oxidoreductase (HAO) constructed based on protein sequences using the neighbor-joining method. Sequences from this study are shown in red (**A**) sample QL-H; (**B**) sample SR-W; (**C**) sample QL-V. Other representative HAO from the protein database and GenBank accession numbers are BAW79844 (*Candidatus Nitrosoglobus terrae*), CUS31385 (*Candidatus Nitrospira nitrosa*), WP_074976104 (*Nitrosospira multiformis*), WP_074793392 (*Nitrosospira briensis*), SEN74695 (*Nitrosomonas marina*), WP_074907499 (*Nitrosomonas communis*), SDZ06451 (*Nitrosomonas halophila*), U04053 (*Nitrosomonas europaea*), WP_074929561 (*Nitrosomonas eutropha*), WP_074202572 (*Nitrosomonas cryotolerans*), SEG22801 (*Nitrosomonas ureae*), SDX60482 (*Nitrosomonas oligotropha*).

The 79 Hao gene sequences were classified based on previous studies by comparing the amino acid sequences of AOB Hao enzymes [20]. As shown in Figure 2A, a phylogenetic analysis of the Hao sequences with the neighbor-joining method showed that the QL-H sample bacterial Hao proteins were affiliated to the *Nitrosomonas* cluster, whereas no sequence from the *Nitrosospira* lineage, which dominates high-nitrogen environments, was observed [21]. All of the retrieved QL-H Hao sequences strongly belonged to the *N. oligotropha* cluster and shared between 89.34–99.63% identity. Only one sequence (Hhao-A45) clustered with Hao of *N. ureae*, with 91.54% amino acid sequence identity. However, 17 of the 25 Hao proteins from the QL-V sample were most affiliated with the *N. oligotropha* cluster and shared 91.18–96.63% amino acid identity; five proteins fell between the *N. communis* and *N. halophila* clusters; and three proteins (Vhao-1, Vhao-02, and Vhao-C8) did not match the Hao proteins of these species (Figure 2C). Interestingly, SR-W sample sequences from Sewage River sludge were sporadically affiliated with the subphyla *N. communis*, *N. halophila*, and *N. oligotropha* clusters, but a new cluster accounted for nearly 80% of the cloned sequences and shared 90.07–99.63% identity with the closest environmental cluster, *N. ureae* (Figure 2B).

### 3.4. Diversity of NirS-Dependent Denitrifying Communities

Nitrite reductase reduces nitrite to nitric oxide in the initial step of two distinct dinitrogen-forming reactions: bacterial denitrification and anammox [2]. They include two structurally different but functionally equivalent forms: copper-containing (NirK) and cytochrome $cd_1$-containing (NirS) nitrite reductases [22]. To understand the diversities of the NirK-type and NirS-type denitrifying microbial communities in different environments, the abundances of the *nirK* and *nirS* genes in the three samples were detected and quantified with metagenomic qPCR. A total of 131 partial *nirS* gene sequences were identified, and 57 final unique *nirS* sequences (74 redundant sequences) were recovered from the sediments of the QL-H, QL-V, and SR-W samples (Figure 3). These results indicated high specificity and sufficient coverage by newly designed primers. These *nirS* genes shared 76–99% identity with other *nirS* sequences in the database. The nucleic acids corresponding to the protein sequences shared relatively high similarities, with approximately 73–100% identity, to matched *nirS* sequences in the NCBI database, which were detected in a variety of environmental samples [8,23].

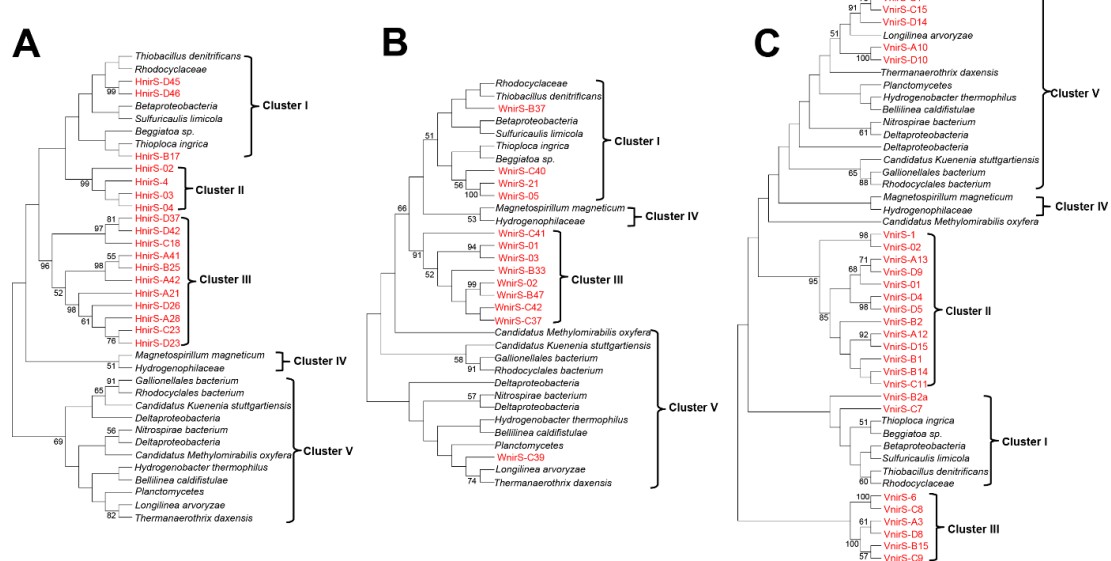

**Figure 3.** Analysis of nitrite reductase (nirS) gene sequences constructed based on protein sequences using the neighbor-joining method. Sequences from this study are shown in red (**A**) sample QL-H; (**B**) sample QL-V; (**C**) sample SR-W. Other representative nirS from the protein database and GenBank accession numbers are OIO76922 (*Hydrogenophilaceae*), WP_011386514 (*Magnetospirillum magneticum*), OFZ68978 (*Betaproteobacteria*), BAV32586 (*Sulfuricaulis limicola*), WP_011310590 (*Thiobacillus denitrificans*), OQY70110 (*Rhodocyclaceae*), OQY57414 (*Beggiatoa* sp.), WP_045480144 (*Thioploca ingrica*), BAE45629 (*Hydrogenobacter thermophilus*), OGW13923 (*Nitrospirae bacterium*), OGP39028 (*Deltaproteobacteria bacterium*), OHB90560 (*Planctomycetes*), WP_054521542 (*Thermanaerothrix daxensis*), KPL78197 (*Bellilinea caldifistulae*), WP_075072401 (*Longilinea arvoryzae*), OGP16868 (*Deltaproteobacteria*), CBE69462 (*Candidatus Methylomirabilis oxyfera*), CAJ74898 (*Candidatus Kuenenia stuttgartiensis*), OGT00430 (*Gallionellales bacterium*), and OHC62023 (*Rhodocyclales bacterium*).

### 3.5. Diversity of NirK-Dependent Denitrifying Communities

Partial *nirK* gene fragments (ca. 500 b.p.) from the three sample libraries and 150 clones were sequenced. A BLAST analysis revealed that only 31 (20%) of the 150 sequences showed homology to known copper-containing nitrite reductase (*nirK*) sequences, with 63–96% nucleotide identity (Figure 4). The deduced amino acid sequences shared 56–99% identity with NCBI NirK sequences. Consistent with a previous classification [24], the phylogenetic tree grouped all the *nirK* sequences into five clusters, and predominantly (86.67%) within the AOB cluster. Interestingly, only nine clones from sample QL-H fell into *nirK*, and six clones belonged to one unique protein. Some clones retrieved

from QL-V and SR-W were grouped with denitrifying bacterial cluster 2, and a single clone from QL-H was grouped into the ammonia-oxidizing archaea (AOA) cluster. These results indicate that bacterial population structures and diversity based on the copper-containing NirK protein differ according to the nitrogen source, which may reflect the occurrence of different 'ecotypes'. Furthermore, a metagenomic qPCR analysis was used to confirm that the copy numbers of *nirS* retrieved from QL-H, QL-V, and SR-W were $1.15 \times 10^5$, $1.76 \times 10^5$, and $7.15 \times 10^4$ gene copies per ng of DNA, respectively, and those of *nirK* were $3.67 \times 10^3$, $8.70 \times 10^3$, and $1.04 \times 10^4$ gene copies per ng of DNA, respectively (Figure 5). The abundances of *nirS* relative to the overall bacterial population were 0.54%, 0.44%, and 0.13%, respectively (Figure 6), and those of *nirK* were 0.017%, 0.022%, and 0.018%, respectively. These results indicate that *nirS* was more abundant than *nirK* in all three samples, which is similar to the results observed in some wastewaters [12]. However, the relative abundances of *nirK* and *nirS* were clearly significantly lower than those in previously reported samples [8]. These differences could be attributable to the wastewater treatment plants having higher carbon and nitrogen concentrations than our three samples and involving treatment processes that benefit denitrifiers, although our samples were also clearly from habitats that benefit denitrifiers.

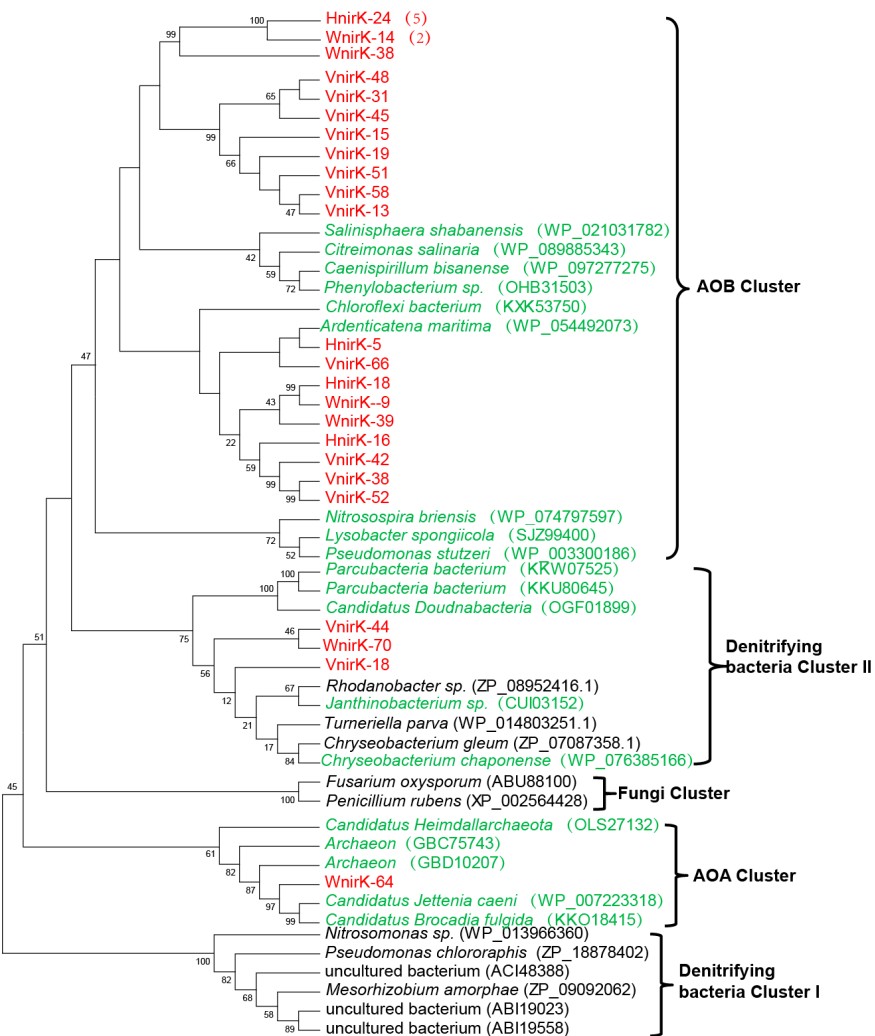

**Figure 4.** Phylogenetic analysis of *NirK* gene products (partial, 140 amino acids) from sample QL-H (HnirK), sample QL-V (VnirK), and sample SR-W (WnirK) bacteria (in red). The 19 candidate *NirK* sequences (in green) were used for designing the *NirS* gene degenerate primer pair. Database sequences are shown in black, with GenBank accession numbers in parentheses. Roman numerals refer to clusters discussed in the text. Bootstrap values (50%) for 1000 replicates are shown at the branch points.

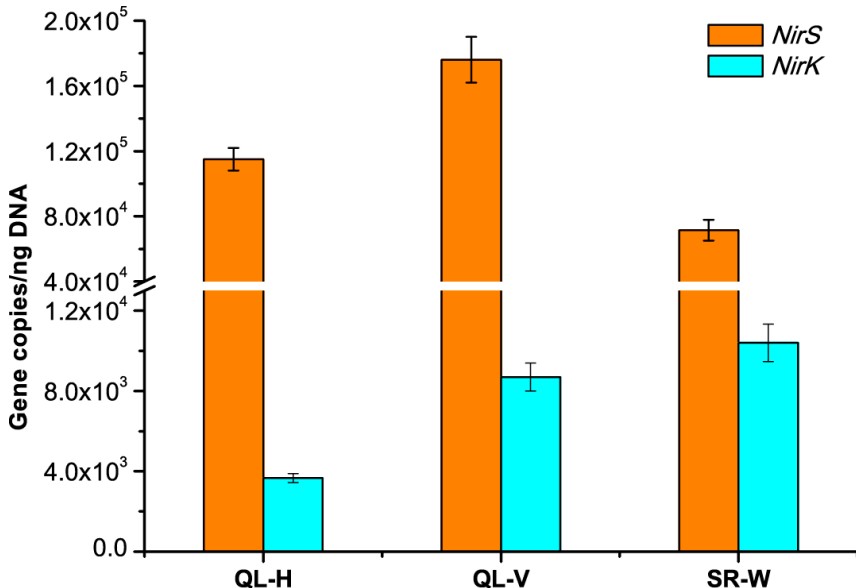

**Figure 5.** PCR measurements for *NirS* and *NirK* gene copy numbers expressed per nanogram of total DNA. Mean values and standard deviations were calculated according to the triplicate assay within a single qPCR setup.

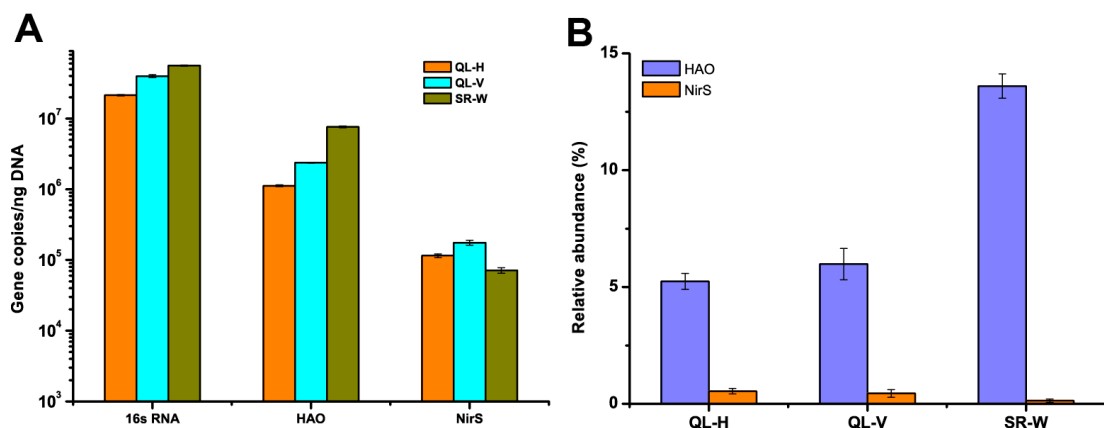

**Figure 6.** Abundance of bacterial 16S rRNA, *HAO,* and *NirS* genes in different sediment samples. (**A**) Copy numbers and (**B**) the relative abundance of HAO and NirS genes are represented as a percentage of their 16S rRNA gene abundance. Mean values and standard deviations were calculated according to the triplicate assay within a single qPCR setup.

### 3.6. Abundances of Nitrifying and Denitrifying Bacterial Communities

Figure 6A shows the qPCR results for the 16S rDNA, *hao*, and *nirS* genes in the three samples. In sludge sample SR-W, with high ammonia and organic contaminant levels, the abundance of 16S rDNA was approximately $5.61 \times 10^7$ copies/ng DNA, whereas in samples QL-H and QL-V, which were from low-ammonia and low-COD environments, the abundances of 16S rDNA were relatively lower (approximately $2.14 \times 10^7$ and $3.98 \times 10^7$ copies/ng DNA, respectively). It indicates that the SR-W sample had a richer microbial habitat than the others. These differences may be attributable to differences in the compositions of soluble substances and the levels of eutrophication. For example, the SR-W sample had higher levels of carbon, nitrogen, and organic matter than the QL-H and QL-V samples from the lake, and dissolved organic matter is known to be an important factor influencing bacterial community structures [25]. The abundances of the *hao* and *nirS* genes of denitrifying bacteria varied significantly among the environmental samples. In sample SR-W, the abundance of the *nirS*

gene was below the limit of detection (about $7.15 \times 10^4$ copies/ng DNA), whereas in samples QL-H and QL-V, which were taken from high-DO low-ammonium environments, the levels of the *nirS* gene were $1.15 \times 10^5$ and $1.76 \times 10^5$ copies/ng DNA, respectively (Figure 6A). Surprisingly, although there were fewer denitrifying bacteria in sludge sample SR-W, the abundance of *hao* genes was significantly higher (three to sixfold) than in the QL-V and QL-H samples ($7.63 \times 10^6$, $2.38 \times 10^6$, and $1.12 \times 10^6$ copies/ng DNA, respectively).

## 4. Discussion

In this study, gene-specific metagenomic PCR and qPCR amplification were used to analyze the environmental factors that influence the microbial community and nitrogen nutrition-related bacterium from three environmental sediment samples with distinct nitrogen source gradients.

Phylogenetic analysis did not show distinct distribution differences in the communities among the nutrient-associated environments. Thus, we further analyzed the overall compositions and distribution of bacterial subdivisions. For example, in sample QL-H, the Proteobacteria were significantly affiliated with the Gammaproteobacteria (21.21%) and Betaproteobacteria (18.19%) clusters, whereas in the SR-W sample, they were more commonly associated with the Betaproteobacteria clusters, which were distributed on two distinct phylogenetic branches, and only one clone was clustered within the Alphaproteobacteria (Figure 1). In contrast, Proteobacteria was not the most abundant phylum in sample QL-V, and all the members were affiliated with the Betaproteobacteria class on distinct phylogenetic branches (Figure 1). The native composition of the microbial community in the QL-H sample was dominated by uncultured bacteria (39.30%) and the Cytophagales-Sphingobacteria group (18.18%). Interestingly, the community diversity in sample QL-V, which was collected in a neighboring village, displayed no relatively concentrated phyla, and uncultured bacteria dominated the clone library. Few clones from the sample QL-V sequences were affiliated with *Aquabacterium*, *Enterovibrio*, or Verrucomicrobia. The sequences from the SR-W sample dataset were distinctly different from those of the other samples, and were assigned to *Aquabacterium* (25%), Betaproteobacteria (25%), Bacteroidetes, and uncultured bacteria. Members of the beta class of the Proteobacteria were well represented under all the environmental conditions, whereas the abundances of the other classes differed significantly. The predominance of Proteobacteria in water samples of different quality suggests that the distribution of structurally simple prokaryotic consortia along microscale biogeochemical gradients is a unique and effective strategy by which bacteria meet their requirements for growth in otherwise inhospitable environments [26]. We also observed a clear difference in the community compositions in the SR-W and QL-H samples, which varied according to their levels of inorganic nitrogen. The genus *Aquabacterium* was overrepresented in the high ammonia–nitrogen water because they have an efficient nitrate-dependent Fe(II) oxidation capacity, using nitrate as an electron acceptor [27]. Overall, the shifts in community composition along the gradients of dissolved organic carbon (DOC) and inorganic contaminants are similar to the patterns of growth efficiency and growth rates [28]. These results suggest that bacterial growth efficiency, abundance, and community composition are presumably constrained by the substrate concentrations of nitrogen compounds in most natural waters, with the possible exception of the most oligotrophic environments [29].

Functional biomarkers related to the physiology of specific bacteria, such as genes encoding unique enzymes involved in their specific biochemical metabolism [30], could offer a better alternative for exploring the diversity and population characteristics of bacterial communities. Furthermore, all the Hao proteins of the AOB clustered with *Nitrosomonas*, whereas no sequence clustered with the *Nitrosospira* lineage (Figure 2). Similar studies have reported that *Nitrosomonas* predominates in tannery wastewater treatment plants [10–12]. However, unlike the predominance of *N. europaea* in various copiotrophic samples, such as laboratory-scale anaerobic ammonium-oxidizing reactors [31], full-scale modified Ludzack–Ettinger process wastewater treatment plants [11], and pilot-scale batch nitrifying reactors [32], none of our recovered clones clustered with *N. europaea*. Instead, *N. oligotropha* and *N. ureae* dominated our three samples. Previous studies have demonstrated that the apparent affinity of

*N. oligotropha*-like AOB for low ammonium may allow them to outcompete the more copiotrophic *N. europaea*-like AOB strains [33]. In this work, it is clear that the Hao-dependent AOB species differed in the three water samples in parallel with the dramatic changes in ammonia nitrogen (Table 1). The three distinct ammonium-graduated sites QL-H (undetectable ammonium), QL-V (16.34 mg/L), and SR-W (93.43 mg/L) AOB were approximately affiliated to *N. ureae* with 2%, 13%, and 80%, respectively. The high concentration of ammonia in the influent of SR-W may have contributed to the predominance of clones related to the *N. ureae* lineage, which was consistent with previous studies that the *N. ureae* cluster was the predominant AOB, was commonly found in environments rich in ammonium [34], and can tolerate up to 200 mM of ammonium [35]. In contrast, *N. oligotropha* already dominated in the AOB of the QL-H sample (>90%), and also made up more than 65% in the QL-V sample and small amounts (~8%) in the RS-W sample. The *N. oligotropha* of AOB species are often found in freshwater due to its lower the *Ks* and *Km* values for ammonia by an order of magnitude more than *N. europaea*, *N. eutropha*, or their close relatives [35], allowing them to grow at low ammonium concentrations [36]. Meanwhile, *N. oligotropha* can also adapt to grow at higher ammonium availability, but with a longer lag phase [36]. Our data show that in oligotrophic environments, carbon and nitrogen are probably important determinants of bacterial community composition and structure, and possibly shifting toward supporting microbial groups. Therefore, the distinct and characteristic *N. oligotropha* population structures in the three environments can be attributed to the sharp differences in the nitrogen concentrations, particularly because in oligotrophic environments, representatives of *N. oligotropha* may outcompete other species.

The newly identified NirS sequences were predominantly affiliated with unknown clusters II and III, and occasionally within unknown cluster I, and the distributions of these subclusters clearly differed. The sequences recovered from the oligotrophic environment of QL-H fell in unknown cluster II (Figure 3A), sharing 75–89% protein sequence identity with other reported sequences. This cluster consisted of four unique NirS sequences and 13 clones that were most closely related to the class Gammaproteobacteria, including the family Xanthomonadaceae (80%) and *Marinobacter* spp. (79%). A previous study demonstrated that members of Xanthomonadaceae could function as autotrophic denitrifiers in a pyrite-fed denitrifying reactor [37], and the *Marinobacter* NirS protein has an important role as a biological marker of microbial degradation in aquatic environments [38]. The grouping of the QL-H sequences with unique cluster II appears to be closely associated with the oligotrophic status of the lake environment, although they formed a new clade within the dominant group. This is consistent with the 16S rRNA results, which showed that the Gammaproteobacteria had high specificity and were highly abundant in the QL-H sample, which was collected from an environment with low ammonium concentration (Figure 1). This new phylogenetic group has rarely been detected in other high ammonia–nitrogen aquatic environments or lakes, and was never retrieved in previous studies of Lake Bourget. Our high success rate with metagenomic PCR screening indicates that this oligotrophic lake is a unique reservoir of diverse and novel nitrite reductase family members, which may represent novel denitrifying bacteria with key roles in limnic biogeochemical nitrogen cycling. In contrast, the sequences recovered from SR-W were predominantly affiliated with unknown clusters I (30.43%) and III (67.39%) (Figure 3B), sharing 89–100% protein sequence identity with previously reported sequences. A single clone grouped in unknown cluster V, sharing 83% protein sequence identity with *Chloroflexi bacterium* [39], which is an autotrophic denitrifier whose performance seems to benefit from nitrate removal. Similarly, members of unique cluster III were also observed in the QL-V sample (20%), together with members of the two dominant unique clusters (I and III) (Figure 3C). The sequences in cluster II shared 73–88% protein sequence identity with sequences retrieved from activated sludge in municipal wastewater and coastal wetlands [9]. Five clones were grouped in cluster V, sharing 77–89% protein sequence identity with sequences retrieved from terrestrial subsurface sediments [40].

These *nir*-based phylogenetic trees revealed important patterns and distributions in the denitrifying community in three different environments providing different nitrogen sources (Figures 3 and 4). The majority of *nirS* clones were found in subclusters within the major clusters formed by unknown

denitrifying bacteria, and did not cluster with any known denitrifying bacteria. In contrast, the clones retrieved from the higher-nitrogen QH-V and SR-W samples clustered with more known denitrifying bacteria than those from QL-H. The distribution and diversity of *nirK*-dependent denitrifiers in the various habitats were similar to those of the *nirS*-dependent denitrifiers. This indicates that the three sediments contained unique denitrifiers, which were unknown among previously reported denitrifiers. These results suggest that the distributions and abundances of some denitrifying bacteria correlated with the bioavailability of nitrogen and carbon, whereas the distributions of others (e.g., *nirS* clusters I and V, and *nirK* cluster AOB) correlated negatively with the carbon and nitrogen sources. Undeniably, some unique clusters were not detected in the two most extreme environments (the most and least eutrophic) from which our samples were drawn, suggesting that they may only occur at very low levels. Previous studies have shown that long-term supplementation with organic fertilizer or the addition of root-derived carbon only slightly affected the denitrifier community structure [1]. Therefore, the distribution of denitrifier clusters may reflect their adaptation to the various intermediate or hybrid environmental conditions they encounter [23,41].

Notably, the qPCR analysis showed that the *hao* gene abundances in all three samples were significantly higher than the *nirS* gene abundances by orders of magnitude. Compared with the 16S rDNA detected in the environmental habitats, the relative abundances of the *hao* gene in samples QL-H, QL-V, and SR-W were 5.23%, 5.97%, and 13.60%, respectively, whereas the relative abundances of the *nirS* gene were 0.54%, 0.44%, and 0.13%, respectively (Figure 6B). The highest abundance of the *hao* gene was detected in sample SR-W from a river containing domestic sewage, where the higher ammonia nitrogen and eutrophication caused nitrifying bacteria to be the dominant bacteria. Notably, the numbers of denitrifying bacterial *nirS* genes were lower than the free-living ones at the closest-to-lake site. Previous studies have shown that total inorganic nitrogen, ammonium, and the organic content are important factors affecting community structures, denitrification rates, and Nir abundance of environmental microbes [25]. With the high concentration of ammonium in the SR-W sample, the accumulation of nitrate and nitrite was significantly higher than in the samples of lake water. Nitrate and nitrite also strongly affect denitrification richness [42], which is consistent with the results for the n-damo enrichments [43]. Environmental factors correlated differently and even oppositely with the 16S rDNA and denitrifier-specific sequences in the three samples, indicating that their influence on the nitrifying and denitrifying community structures is very complex.

Together, we detected nitrifying and denitrifying bacteria in three complex and contrasting environments with gene-specific metagenomic PCR amplification of the unique functional marker genes *hao*, *nirS*, and *nirK*. This provided comprehensive evidence that the microbial community structures of nitrifiers and denitrifiers are influenced by the environmental nitrogen status. The *hao* genes of the majority of nitrifying bacteria retrieved and the *nirK* and *nirS* genes of the denitrifying bacteria were affiliated with those of different genera of uncultured bacteria, but distantly related to the genes of known nitrifiers and denitrifiers, respectively. The high diversities and abundances observed in the Hao and Nir phylogenies suggest high interspecies variety in the three different environments (which differed sharply in their nitrogen and organic carbon status), which caused pronounced shifts in the nitrifying and denitrifying populations. The characteristic nitrifier and denitrifier population structures differed according to the nitrogen source, which was reflected in the predominant potential nitrifying and denitrifying genera present, and especially in the many redundant clones that were present, which may contribute greatly to the nitrogen metabolism in these unique environments. These results extend our knowledge of the biology and diversity of nitrifiers and denitrifiers in natural lakes and polluted rivers, and may have practical utility in the efficient restoration of rivers and lakes. However, the high diversity of marker genes and the changing patterns of bacterial dominance do not really reflect their contribution to nitrogen cycling. Novel experimental approaches such as transcriptome, proteomics, and metabolomics [44] are required to detect active nitrifiers and denitrifiers, to better understand their roles and the mechanisms they use in nitrogen cycling and environmental bioremediation.

**Author Contributions:** P.J. and Z.Z. wrote the manuscript. Sample collection, processing and analysis were carried out by R.Z., Y.C., J.Q. and Z.C.

**Funding:** This research was supported by the National Natural Science Foundation of China (Grant 31700078 and Grant 31270724), the Scientific Research Foundation for Talent program of Zhejiang Agricultural and Forestry University (W20170029), Distinguished Scholars of Zhejiang Agricultural and Forestry University (2014FR064).

**Conflicts of Interest:** The authors declare no competing financial interests.

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
