# Peer review of "Insights into the Variations of Hao-Dependent Nitrifying and Nir-Dependent Denitrifying Microbial Communities in Ammonium-Graduated Lake Environments"

_applsci, doi:10.3390/app9163229_

Round 1
Reviewer 1 Report
The authors have successfully addressed the reviewer's concerns and the manuscript can be accepted for publication.
Author Response
Cover letter for revision
Dear editor,
This is the revised manuscript (manuscript ID:applsci-557066) by Ruojin Zhao entitled “Insights into the variations of Hao-dependent nitrifying and denitrifying microbial communities in ammonium-graduated lake environments”. We sincerely appreciate the editor and reviewers’ comments and suggestions for our manuscript. According to these comments and suggestions, we have carefully revised the manuscript (The changes were marked in red), carefully answered and explained all the queries and suggestions by point-by-point in the submitted responses, which we hope meet with approval.
We shall look forward to hearing from you at your earliest convenience. Thank you very much.
Sincerely yours,
Dr. Peng Jin.
E-mail address: jinpeng@zafu.edu.cn.
The College of Agricultural and Food Sciences, Zhejiang A&F University, Hangzhou, Zhejiang, China
Response to Reviewer 1 Comments
Point 1:The authors have successfully addressed the reviewer's concerns and the manuscript can be accepted for publication.
Response 1:we sincerely appreciate the reviewer 1’ positive comments, suggestions and time on our manuscript. Those comments are all valuable and very helpful for revising and improving our paper, as well as the important guiding significance to our researches.
Reviewer 2 Report
General comments
- This is a useful study that will be valuable to the readers of this journal.
- Minor grammatical errors in the manuscript should be corrected.
Specific Comments:
Materials and Methods:
- Page 3/15, line 93: A reference should be provided for the Winkler method.
Results:
- Table 1 should be presented in the Results section 3.1.
- Page 4/25, line 165: What kind of statistical analysis was conducted? This should be briefly discussed in the Materials and Methods section.
- The authors did not discuss the impact of dissolved oxygen concentration and oxidation-reduction potential (ORP) in the environmental sediment samples of river and lake on the diversity of microbial population. This should be discussed.
Author Response
Cover letter for revision
Dear editor,
This is the revised manuscript (manuscript ID:applsci-557066) by Ruojin Zhao entitled “Insights into the variations of Hao-dependent nitrifying and denitrifying microbial communities in ammonium-graduated lake environments”. We sincerely appreciate the editor and reviewers’ comments and suggestions for our manuscript. According to these comments and suggestions, we have carefully revised the manuscript (The changes were marked in red), carefully answered and explained all the queries and suggestions by point-by-point in the submitted responses, which we hope meet with approval.
According to suggestions of reviewers, the changes of the manuscript are briefly as follows:
Line 91, added areference [18] for the Winkler method; Line 151-154,Table 1 was placed here from Line 75-77; and added the DO values of three environmental samples; Line 168 the incorrect description “statistical analysis” wasrevised it with “phylogenetic analysis”.
We shall look forward to hearing from you at your earliest convenience. Thank you very much.
Sincerely yours,
Dr. Peng Jin.
E-mail address: jinpeng@zafu.edu.cn.
The College of Agricultural and Food Sciences, Zhejiang A&F University, Hangzhou, Zhejiang, China
Response to Reviewer 2 Comments
Point 1:General comments
This is a useful study that will be valuable to the readers of this journal.
Minor grammatical errors in the manuscript should be corrected.
Response 1:Firstly, we sincerely appreciate the reviewer 2’ valuable suggestions and positive evaluation on our work. According to these comments and suggestions, we have carefully revised the manuscript which we hope meet with approval.
Point 2:Materials and Methods: Page 3/15, line 93: A reference should be provided for the Winkler method.
Response 2:Thank the reviewer 2 for this good suggestion. Accordingly, we have added the reference for the Winkler method in the manuscript.
Point 3:Results: Table 1 should be presented in the Results section 3.1.
Response 3:Thank the reviewer 2 for the good suggestion. As suggestion, we have put Table 1 in the Results section 3.1.
Point 4: Page 4/25, line 165: What kind of statistical analysis was conducted? This should be briefly discussed in the Materials and Methods section.
Response 4:Thank the reviewer 2 for the careful observation. We are very sorry for our incorrect description about statistical analysis. Accordingly, we have revised it with “phylogenetic analysis” in the manuscript.
Point 5:The authors did not discuss the impact of dissolved oxygen concentration and oxidation-reduction potential (ORP) in the environmental sediment samples of river and lake on the diversity of microbial population. This should be discussed.
Response 5:Thank the reviewer 2 for this good suggestion. In fact, according to our DO measurements, DO did not exhibit a gradient or distinct trends, so we did not conduct in-depth analysis on DO and ORP. According to the good suggestion, we have added the data of DO in the revised manuscript. Of course, we also sincerely appreciate this good suggestion. In the further work, we will work on the effects of DO and ORP on the diversity of microbial population in environmental samples.
This manuscript is a resubmission of an earlier submission. The following is a list of the peer review reports and author responses from that submission.
Round 1
Reviewer 1 Report
Major comments
The manuscript describes just the cloning results of 16S rRNA gene and nitrifying and denitrifying related genes. This is just a snap shot of microbial communities and there is no novelty. There are lot of related publications. So authors should describe the novelty and importance of current study more in detail and clearly.
Specific comments
Figure 1: Please include the other isolated species as a reference. Current phylogenetic tree is difficult to understand.
Materials and Methods, section 2.3: Which primers were used for qPCR? Same as cloning?
Please describe the coverage of newly designed primers and also compare the previous primers.
Author Response
Response to Reviewer 1 Comments
Point 1:The manuscript describes just the cloning results of 16S rRNA gene and nitrifying and denitrifying related genes. This is just a snap shot of microbial communities and there is no novelty. There are lot of related publications. So authors should describe the novelty and importance of current study more in detail and clearly.
Response 1:Firstly, we sincerely appreciate the reviewer 1’s valuable suggestions and evaluation on our work. We used gene-specific metagenomic PCR and qPCR approaches to explore the effects of ammonium-graduated lake environments on nitrifying and denitrifying microbial communities. This study provides valuable comparative insights into the abundance, diversity, and compositions of nitrifying and denitrifying microbial populations, which varied differed under the nitrogen source gradients. These results extend our knowledge of the biology and diversity of nitrifiers and denitrifiers in natural lakes and polluted rivers, and may have practical utility in the efficient restoration of rivers and lakes.
According to these comments and suggestions, we have carefully revised and added the corresponding results in the revised manuscript which we hope meet with approval. The detailed responses to the comments are shown below.
Point 2:Figure 1: Please include the other isolated species as a reference. Current phylogenetic tree is difficult to understand.
Response 2:Thank the reviewer 1 for this good suggestion. In fact, we have confirmed all the clone sequences to their bacterial species by NCBI online Blast. So here we (figure 1) did not construct the phylogenetic tree with other known reference sequences, but only for cluster analysis of all sample sequences more directly.
Point 3:Materials and Methods, section 2.3: Which primers were used for qPCR? Same as cloning?
Response 3:Thank the reviewer 1 for the careful observation. Primers used for quantification were the same as their gene-specific degenerate primers of cloning.
Point 4:Please describe the coverage of newly designed primers and also compare the previous primers.
Response 4:Thank the reviewer 1 for this good suggestion. In fact, we have attempted and compared with other reported primers, and found that the PCR amplification effect was not ideal for the metagenomic DNA. We also have designed multiple pairs of primers for these genes and carried out the combinatorial optimization (about 90% coverage). Meantime, we will continue to optimize the specificity of degenerate primers in subsequent studies to obtain the best amplification effect of the experiment.
Reviewer 2 Report
The manuscript entitled “Insights into the variations of Hao-dependent nitrifying and Nir-dependent denitrifying microbial communities in ammonium-graduated lake environments” is an interesting research paper that aims to examine the nitrifying and denitrifying microbial communities through gene-specific PCR techniques. Although, the study focuses on three specific marker genes (e.g. hao for nitrifiers, and nirS/K for denitrifiers) the collected data are interesting and well-presented. Moreover, the authors provide an extensive discussion section that is considered of high significance. The manuscript has some minor points that need to be addressed by the authors:
a. Lines 17-18: Please consider rephrasing “within the ammonium…form a gradient.”; the meaning is unclear.
b. Lines 72: Please replace full-stop with comma.
c. Table 1: Does these values refer to the water or the sediment data?
d. Lines 88-94: Please make clear whether you collected water samples apart from sludge samples or only sludge samples.
e. Line 209: Why do the authors mention the SW-R sample as “sewage sludge”?
f. Figure 2: It seems that there is a mistake in the figure markings; The B phylogenetic tree seems to refer to SR-W sample and the C to the QL-V sample. Please, check and correct, accordingly.
g. Figure 5: The use of two scales at the y-axis is misleading. Please, consider using one scale for both NirS and NirK data. The use of logarithmic scale can be also considered.
h. Line 284: Please, replace “low” with “lower”.
i. Line 356: The phrase “of magnitude…relatives do [33]” is unclear.
Author Response
Response to Reviewer 2 Comments
Point 1:The manuscript entitled “Insights into the variations of Hao-dependent nitrifying and Nir-dependent denitrifying microbial communities in ammonium-graduated lake environments” is an interesting research paper that aims to examine the nitrifying and denitrifying microbial communities through gene-specific PCR techniques. Although, the study focuses on three specific marker genes (e.g. hao for nitrifiers, and nirS/K for denitrifiers) the collected data are interesting and well-presented. Moreover, the authors provide an extensive discussion section that is considered of high significance. The manuscript has some minor points that need to be addressed by the authors:
Response 1: Firstly, we sincerely appreciate the reviewer 2’s valuable suggestions and positive evaluation on our work. According to these comments and suggestions, we have carefully revised and added the corresponding results in the revised manuscript which we hope meet with approval. The detailed responses to the comments are shown below.
Point 2:Lines 17-18: Please consider rephrasing “within the ammonium…form a gradient.”; the meaning is unclear.
Response 2:Thank the reviewer 2 for this good suggestion. We have rewritten this statement to make it clearer.
Point 3:Lines 72: Please replace full-stop with comma.
Response 3:Thank the reviewer 2 for the careful observation. We have revised the full-stop with comma.
Point 4:Table 1: Does these values refer to the water or the sediment data?
Response 4:Thank the reviewer 2 for the careful observation and good suggestion. All samples were collected from the surface sediments (5 cm) at a water depth of 30–50 cm. So these values referred to the sediment data.
Point 5:Lines 88-94: Please make clear whether you collected water samples apart from sludge samples or only sludge samples.
Response 5:Thank the reviewer 2 for the careful observation and good suggestion. We are very sorry for the ambiguous description, and have corrected it in the revised manuscript. All the samples were collected from sludge.
Point 6:Line 209: Why do the authors mention the SW-R sample as “sewage sludge”?
Response 6:Thank the reviewer 2 for the careful observation and good suggestion. We have revised it with “sewage river sludge”. The mention of the SW-R sample as “sewage river sludge” is mainly to emphasize its environmental conditions for the comparison and discussion.
Point 7:Figure 2: It seems that there is a mistake in the figure markings; The B phylogenetic tree seems to refer to SR-W sample and the C to the QL-V sample. Please, check and correct, accordingly.
Response 7:Thank the reviewer 2 for the careful observation and good suggestion. We have corrected it in the revised manuscript.
Point 8:Figure 5: The use of two scales at the y-axis is misleading. Please, consider using one scale for both NirS and NirK data. The use of logarithmic scale can be also considered.
Response 8:Thank the reviewer 2 for this good suggestion. As suggestion, we have carefully revised the Figure 5 in the new manuscript. We tried to use the logarithmic scale, but the Y-axis bar was too large to make the graph ideal. So we used the one scale (gene copies/ng DNA) for both NirS and NirK data as shown in the revised manuscript.
Point 9:Line 284: Please, replace “low” with “lower”.
Response 9:Thank the reviewer 2 for this good suggestion. Accordingly, we have revised it in the revised mmanuscript.
Point 10:Line 356: The phrase “of magnitude…relatives do [33]” is unclear.
Response 10:Thank the reviewer 2 for the careful observation and good suggestion. We have deleted the word “do”, so it was clearer.
Round 2
Reviewer 1 Report
Authors have provided the bare-minimum in response to reviewers' comments, and in-depth analysis or significant revision that was requested is still missing. The revised submission still does not represent a sufficient advance in our understanding.